# Application of artificial neural networks and genetic algorithm to predict and optimize greenhouse banana fruit yield through nitrogen, potassium and magnesium

**Mahmoud Reza Ramezanpour[1], Mostafa Farajpour[2]***

**1** Soil and Water Research Department, Mazandaran Agricultural and Natural Resources Research and Education Center, AREEO, Sari, Iran, **2** Crop and Horticultural Science Research Department, Mazandaran Agricultural and Natural Resources Research and Education Center, AREEO, Sari, Iran

* Farajpour_m@ut.ac.ir

**Data Availability Statement:** All relevant data are within the paper.

## Abstract

The excess of the chemical fertilizers not only causes the environmental pollution but also has many deteriorating effects including global warming and alteration of soil microbial diversity. In conventional researches, chemical fertilizers and their concentrations are selected based on the knowledge of experts involved in the projects, which this kind of models are usually subjective. Therefore, the present study aimed to introduce the optimal concentrations of three macro elements including nitrogen (0, 100, and 200 g), potassium (0, 100, 200, and 300 g), and magnesium (0, 50, and 100 g) on fruit yield (FY), fruit length (FL), and number of rows per spike (NRPS) of greenhouse banana using analysis of variance (ANOVA) followed by post hoc LSD test and two well-known artificial neural networks (ANNs) including multilayer perceptron (MLP) and generalized regression neural network (GRNN). According to the results of ANOVA, the highest mean value of the FY was obtained with 200 g of N, 300 g of K, and 50 g of Mg. Based on the results of the present study, the both ANNs models had high predictive accuracy ($R^2 = 0.66$–0.99) in the both training and testing data for the FY, FL, and NRPS. However, the GRNN model had better performance than MLP model for modeling and predicting the three characters of greenhouse banana. Therefore, genetic algorithm (GA) was subjected to the GRNN model in order to find the optimal amounts of N, K, and Mg for achieving the high amounts of the FY, FL, and NRPS. The GRNN-GA hybrid model confirmed that high yield of the plant could be achieved by reducing chemical fertilizers including nitrogen, potassium, and magnesium by 65, 44, and 62%, respectively, in compared to traditional method.

## Introduction

After citrus, banana (*Musa sapientum*) with 16% of all fruits products, is the largest fruit product worldwide [1]. Due to different banana uses including traditional, medicinal, and nutritional, the plant is one of the most important fruits worldwide. It helps to retain healthy tissues

**Funding:** The authors received no specific funding for this work.

**Competing interests:** The authors have declared that no competing interests exist.

through maintaining Ca, P, and N in the body [2]. The plant fruit helps body to control the blood pressure and sugar and keeps the bones strong, because of the amount of magnesium [3]. Banana is relatively low in calories, however, it is rich in carbohydrates, potassium, and vitamins (C, B6, and A) [4]. In addition, the peels of the plant have various health benefits due to high levels of phenolic component resulted in potential antioxidant and antimicrobial activities [5]. Because of high production of banana (115.74 million tonnes) worldwide, the industry of plant is important for the global industrial agribusiness [6]. The total area of cultivation concerning banana in Iran is an estimated about 4800 ha, which produced about 140,000 tons [7]. Plants during their life cycle, need essential nutrients of two categories, i.e. macronutrients and micronutrients. Fertilization is one of the most important factors for enhancing the growth and production of banana plants due to the plant's high need for nutrients [8]. The order of absorption of macronutrients in the plant includes potassium > nitrogen > calcium > magnesium. Among the macronutrients, banana requires and accumulates high levels of potassium and nitrogen in its tissues [9]. The potassium is require for osmoregulation, stomatal behavior, transpiration, and photosynthesis [10]. However, nitrogen is requiring for the plant growth, photosynthesis, and respiration [11]. Another important macronutrient for banana is Mg, which the deficiency of this element decreased the biomass and root and shoot dry weight ratio of the plant [12]. In many agricultural experiments, to determine the appropriate amount of fertilizer for the plant, several fertilizers are given to the plant at different levels and finally the best treatment is introduced [13–15]. However, there is little use of scientific tools to determine the appropriate amount of several fertilizers for the plant [16]. As there are different interactions among the fertilizers with plants, make it a complex interaction process. Therefore, the conventional statistical approaches including ANOVA followed by a post hoc test such as LSD and Duncan are not enough to study the appropriate amount of several fertilizers for different plants. In conventional researches, the structural design, factors, and their concentrations are selected based on the knowledge of experts involved in the projects, which this kind of models are usually subjective [17]. However, the computational techniques are efficiently construct and optimize fertilizing models. Machine learning (ML) algorithms have been used to predicting and optimizing different complex biology and agriculture systems [18–21]. Among the various ML algorithms, artificial neural networks (ANNs) have been considered as the most well-known models [22]. The effectiveness and reliability of these models in investigating and predicting different fertilizers have been confirmed in different plants such as wheat [23] and oat [24]. Genetic algorithm (GA) is a well-known single objective optimization algorithm, which a hybrid with ANN (ANN-GA) make them a potential hybrid model for modeling and optimizing in different biology and agriculture systems such as soil heavy metals [25], plant tissue culture [19, 26, 27], and adsorption and photocatalysis of a synthesized nanomaterial [28]. However, to our best knowledge there is no any study to apply ANN-GA hybrid model for predicting and optimizing agriculture firtilizers i.e. nitrogen, potassium, magnesium, and etc. The present study aimed to (1) assessment of the effects of nitrogen, potassium, and magnesium on the fruit yield (FY), fruit length (FL), and number of rows per spike (NRPS) in *M. sapientum*, (2) modeling and predicting the plant characters using two known ANNs such as multilayer perceptron (MLP) and radial generalized regression neural network (GRNN), and (3) introduce the optimal concentrations of nitrogen, potassium, and magnesium for the measured plant characters usnig the GA.

## Material and methods

The experiment was under greenhouse conditions in Sari, Mazandaran Province, Iran, on two months old banana (Grand Nain cultivar) during two years. The experiment was a split plot

factorial based on completely randomized design (CRD). The three macronutrient fertilizers were, namely, nitrogen (0, 100 and 200 g), potassium (0, 100, 200, and 300 g), and magnesium (0, 50 and 100 g) for each plant from the sources of ammonium, potassium, and magnesium sulfates, respectively. In addition, the phosphate fertilizer was given at a rate of 50 g per pot. The magnesium sulfate fertilizer was initially added to the pots. However, we divided the amount of nitrogen fertilizer in two levels of 75 and 150 g into three parts and added it to the pots in three stages including starting the experiment, two months and three months after the experiment. Also, the treatments of 100, 200 and 300 g of potassium were divided into four parts which were added into the pots in four stages such as starting the experiment and two, three, and four months after the experiment. The temperature and humidity of greenhouse were maintained at 27 ± 2˚C and 85–90%, respectively. Irrigation was done by drip irrigation with one emitter per pot. In the present study three traits of banana trees including FY, FL, and NRPS, were measured at fruit ripening time.

Finally, the data were used for modeling, predicting, and optimizing FY, FL, and NRPS of greenhouse banana. The step-by-step procedure of the experiment was shown in Fig 1.

## Statistical analysis

**ANOVA.**   The present study was conducted during two years under greenhouse conditions. As there was no significant difference between the data for the two years of the three studied traits, therefore, the mean values of the two years were used for analysis of variance (ANOVA). The ANOVA and the least significant difference (LSD) test were achieved using Statistical Analysis Software (Version 9.1 for Windows; SAS Institute, Cary, NC). The cluster analysis was used to find the relationship among the treatments based on the arithmetic mean method (UPGMA). The cluster and heat map correlation analyses were visualized as a colored

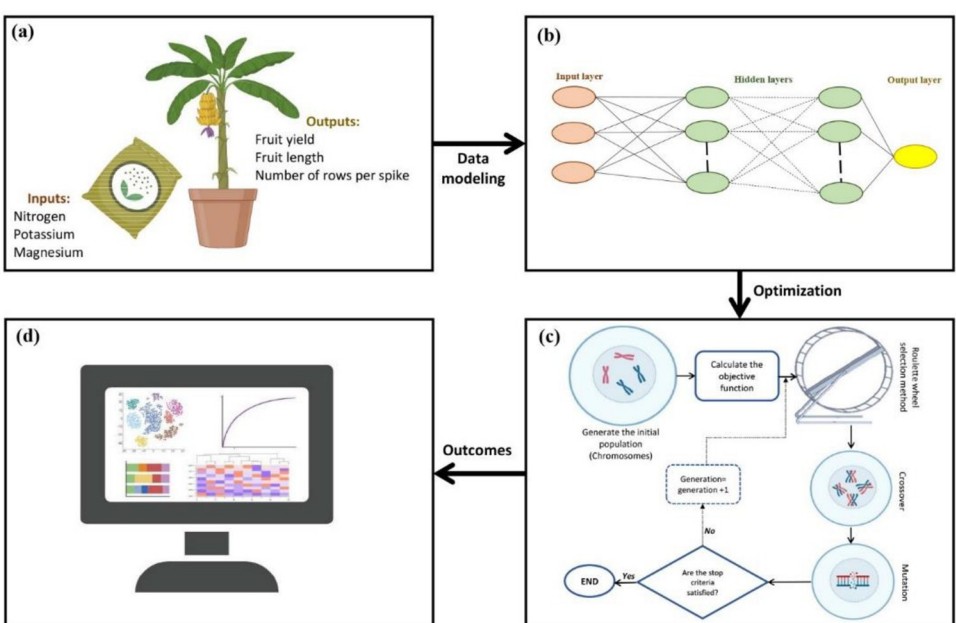

**Fig 1.**  The schematic view of the step-by-step procedure of the present study (a) data collection, (b) modeling the FY, FL, and NRPS based on three input variables including N, K, and Mg using ANNs, (c) optimization process via genetic algorithm and (d) outcomes.

heat map using MetaboAnalyst [29]. Also, the scatter plots were produced by GraphPad Prism software v.7 (GraphPad, La Jolla California USA).

**The ANNs.** The ANNs models including MLP and GRNN were used to model and predict the three measured banana traits. In the present study, Box-Cox transformation and principal component analysis (PCA) were used to normalize and detect outliers the data, respectively. In addaition, to assess the prediction accuracy of the models, five-fold cross-validation were performed. In this study, three fertilizers including nitrogen, potassium, and magnessium were selected as input variables, while three measured banana traits including FY, FL, and NRPS were considered as output variables. Finaly, the two ANNs models were compared with respect to their coefficient of determination (R2), Mean Bias Error (MBE) and Root Mean Square Error (RMSE).

**MLP and GRNN models.** The 3-layer backpropagation MLP is a distributed and parallel model that employs supervised learning for the training set that minimizes the error between the output and input variables through the following function:

$$\text{Error} = 1/\text{n} \sum_{n=1}^{n} (y_s - \hat{y}_s)^2$$

where n is number of observations, and $y_s$ and $\hat{y}_s$ are the sth observed and predicted variables, respectively.

The following equation was implemented to compute the $\hat{y}$ in the MLP model

$$\hat{y} = f[{}^{p}\sum_{j=1} w_j \cdot g \left({}^{k}\sum_{i=1} w_{ji} x_i + w_{j0}\right) + w_o]$$

where p and k are number of neurons in the hidden layer and output variables, respectively. $x_i$ is the ith output variable, $w_j$ shows the weighted input data into the jth hidden neuron, f shows activation function for the output neuron, $w_{ji}$ represents the weight of the direct relationship of input neuron i to the hidden neuron j, $w_{j0}$ shows the bias for node jth, $w_0$ equals the bias connected to the neuron of output, and g represents the activation function for the hidden neuron. In the present study, the optimal neuron number in the hidden layer was detected according to trial and error. Also, the purelin and the tansig were used as the transfer functions of output and hidden layers, respectively. Finally, a Levenberg Marquardt algorithm was used to adjust the bias and weights.

The GRNN model consists of four layers including input, pattern, summation, and output layers. The node in input layer completely enters the node in pattern layer. The output of each neuron in pattern layer is connected to the summation neurons. The unweighted and weighted pattern neuron outputs were calculated using D- and S-summation neurons, respectively. In order to determine the output, following formula is used

$$\hat{y} = \frac{\sum_{i=1}^{n} yi \exp\left(-\frac{D_i^2}{2\sigma^2}\right)}{\sum_{i=1}^{n} \exp\left(-\frac{D_i^2}{2\sigma^2}\right)}$$

$$D_{i^2} = (x - x_i)^T (x - x_i)$$

where σ shows width parameter, $\hat{y}$ represents the mean of all the weighted observed output data, $y^i$ represents the $i^{th}$ output variable, and $D_i^2$ represents a scalar function according to any xi and yi observed data.

**The GA.** In the present study, the GA was employed to optimize the appropriate levels of the three fertilizers to obtain the highest mean values the greenhouse banana characters. The upper and lower bounds were recognized as constraints, and the point with the highest value for FY, FL, and NRPS was considered as the optimal value. The 2-point crossover and the

uniform of mutation functions along with the roulette wheel as a selection function, were used during the optimization process. In order to obtain, the following parameters were set to the crossover rate with a value of 0.7, generation number with 1000, initial population with 200, and mutation rate with 0.04 values. The algorithm codes written by MATLAB (MATLAB 2018) software.

## Results and discussion

### ANOVA followed by post hoc LSD test

According to the results of ANOVA, there were significant differences (P <0.01) among N, K, and Mg and their interactions for FY, FL, and NRPS. According to Wickens and Keppel [30], when the tripartite interaction is significant, then less attention is paid to other effects. Based on the results the highest mean value of the FY (119.1 kg per plant) was obtained with 200 g of N, 300 g K and 50 g of Mg (Table 1). However, the lowest mean values of this character were observed in control, and 50 and 100 g of Mg (without N and K). These results indicated that Mg was not affective to enhanced the FY in compared with N and K. Fratoni et al. [9] showed that the plant yield was not significantly affected by N, K and also their interaction. However, the applications of these two fertilizers for banana trees, are the main step for enhancing both the FY and quality [31]. These facts were confirmed by the results of current study, and Marie-Laure et al. [32] which reported that the applications of N and K in different concentrations enhance the yield of the plant in compared with control. In addition, Islam et al. [33] revealed that applications of 500 g of N, 450 g of K and their interactions (500 g of N and 450 g of K) were affective to enhance the FY in compared with the control. Based on the results of present study, the interaction effects of N and K caused to enhance the FY and in many treatments with increasing the Mg levels, the FY decreased. The interactions among the fertilizers are important determinants of different plant yield, however, the mechanisms involved remain poorly understood. The interactions between N and K enhanced the rice yields by improving canopy performance [34]. The K/Mg ratio is a critical ratio for banana production, and the ratio above 0.7 caused to appear the physiological disorders resulted in reduction in FY [35]. Based on the results, the highest FL was belonged to 200 g of N, and K, and 50 g of Mg treatment. This treatment enhanced the FL by 260% in compared with the control. The increasing in FL by N and K is related to the concentration of these elements and banana variety [32]. The highest mean values of NRPS obtained under 200 g of N and K with 100 and 50 g of Mg. In order to find the relationships between the three measured banana characters including FY, FL, and NRPS, heat-map correlation analysis was performed. The results of this analysis revealed that the three measured characters had significant correlation with each other (Fig 2; P < 0.01). In addition, to find similar fertilizers treatments, cluster analysis was employed. The 36 treatments derived from three fertilizers factors, were classified in five main groups (Fig 3). Based on the results, the highest mean values of the FY, FL, and NRPS were observed in the five treatments in the third group. In all of these five treatments the concentration of N was 200 g and K was 200 or 300 g, however, the three concentrations of Mg were observed. These results revealed that the different concentrations of the three fertilizers were able to enhance the FY, FL, and NRPS. For example, in the highest mean values of the FY, the amounts of K and Mg were variable. These low and high amounts of fertilizers cause to increase the production costs, environmental pollution and adverse effects on the plant. The high level of K can inhibit magnesium uptake and resulted in magnesium deficiency in plants (Tränkner et al., 2018). The addition of the chemical fertilizers not only causes the environmental pollution but also has many deteriorating effects including global warming and alteration of soil microbial diversity [36]. Based on the results of present study, the conventional statistical approaches

**Table 1. The effects of N, K, and Mg on FY, FL, and NRPS in banana.**

| N | K | Mg | FY (kg/plant) | FL (cm) | NRPS |
|---|---|---|---|---|---|
| 0 | 0 | 0 | 42.1 ± 2.52 | 9.63 ± 0.47 | 5.77 ± 0.29 |
| | | 50 | 43.8 ± 2.14 | 10.4 ± 0.47 | 6.23 ± 0.29 |
| | | 100 | 45.25 ± 3.97 | 11.77 ± 0.52 | 6.1 ± 0.2 |
| | 100 | 0 | 52.37 ± 5.76 | 12.27 ± 0.32 | 6.1 ± 0.2 |
| | | 50 | 50.1 ± 2.31 | 12.2 ± 0.32 | 6.3 ± 0 |
| | | 100 | 50.6 ± 1.35 | 12.5 ± 0.35 | 6.57 ± 0.13 |
| | 200 | 0 | 52.9 ± 3.1 | 13.83 ± 0.68 | 6.9 ± 0.2 |
| | | 50 | 53.5 ± 2.02 | 14.33 ± 0.88 | 7.1 ± 0.2 |
| | | 100 | 53.5 ± 1.71 | 13.03 ± 0.52 | 7.2 ± 0.1 |
| | 300 | 0 | 54.4 ± 2.91 | 13.4 ± 0.65 | 8.1 ± 0.2 |
| | | 50 | 56.1 ± 1.57 | 14.8 ± 0.44 | 8.2 ± 0.1 |
| | | 100 | 60.25 ± 2.38 | 14.6 ± 0.4 | 8.43 ± 0.13 |
| 100 | 0 | 0 | 62.1 ± 2.39 | 14.1 ± 0.99 | 8.53 ± 0.39 |
| | | 50 | 61.45 ± 2.68 | 13.93 ± 0.22 | 9 ± 0.17 |
| | | 100 | 63.52 ± 2.79 | 15.2 ± 0.47 | 9.43 ± 0.47 |
| | 100 | 0 | 91.2 ± 12.24 | 21.47 ± 0.64 | 11.33 ± 0.49 |
| | | 50 | 85.44 ± 7.1 | 22 ± 0.32 | 11.43 ± 0.13 |
| | | 100 | 92.4 ± 5.97 | 21.57 ± 0.64 | 11.2 ± 0.1 |
| | 200 | 0 | 100.85 ± 5.09 | 21.2 ± 0.82 | 12 ± 0.17 |
| | | 50 | 102.15 ± 6.07 | 21.27 ± 0.88 | 12.7 ± 0 |
| | | 100 | 83.55 ± 11.14 | 17.47 ± 0.93 | 9.3 ± 0 |
| | 300 | 0 | 82.95 ± 10.95 | 18.67 ± 1.53 | 9.77 ± 0.29 |
| | | 50 | 69 ± 4.46 | 18.4 ± 1.07 | 10.23 ± 0.79 |
| | | 100 | 89.5 ± 11 | 22.13 ± 0.78 | 11.77 ± 0.47 |
| 200 | 0 | 0 | 83.65 ± 2.28 | 17.6 ± 0.98 | 9.23 ± 0.29 |
| | | 50 | 84 ± 6.59 | 17.88 ± 0.9 | 10.3 ± 0 |
| | | 100 | 79.1 ± 2.54 | 18.77 ± 0.41 | 10.1 ± 0.1 |
| | 100 | 0 | 78 ± 1.15 | 19.77 ± 0.85 | 10.7 ± 0 |
| | | 50 | 86.05 ± 1.92 | 19.57 ± 0.45 | 10.77 ± 0.29 |
| | | 100 | 83.7 ± 4.69 | 20.23 ± 0.41 | 11.43 ± 0.47 |
| | 200 | 0 | 98.8 ± 9.13 | 21.9 ± 0.92 | 12.2 ± 0.1 |
| | | 50 | 107.3 ± 6.04 | 25.27 ± 0.38 | 13.77 ± 0.29 |
| | | 100 | 113.4 ± 3.96 | 25.23 ± 0.47 | 14.2 ± 0.1 |
| | 300 | 0 | 116.1 ± 6.26 | 23.47 ± 0.27 | 12.9 ± 0.2 |
| | | 50 | 119.1 ± 6.75 | 24.3 ± 0.56 | 13.23 ± 0.47 |
| | | 100 | 111.5 ± 4.6 | 24 ± 0.21 | 12.77 ± 0.29 |
| LSD | - | - | 10.58 | 1.86 | 0.82 |

Values in each column represent as the averages of three replications ± SEM.

FY: fruit yield, FL: Fruit length, and NRPS: number of rows per spike.

including ANOVA was not enough to study the appropriate amounts of the three fertilizers for the plant. Therefore, it is necessary to optimize the concentration of agricultural fertilizers through computational approaches such as machine learning algorithms.

Therefore, MLP and GRNN models were applied to predict the FY, FL, and NRPS of greenhouse banana according to the three fertilizers. According to the results, the GRNN model showed higher predictive accuracy in the training and testing sets in compared with the MLP

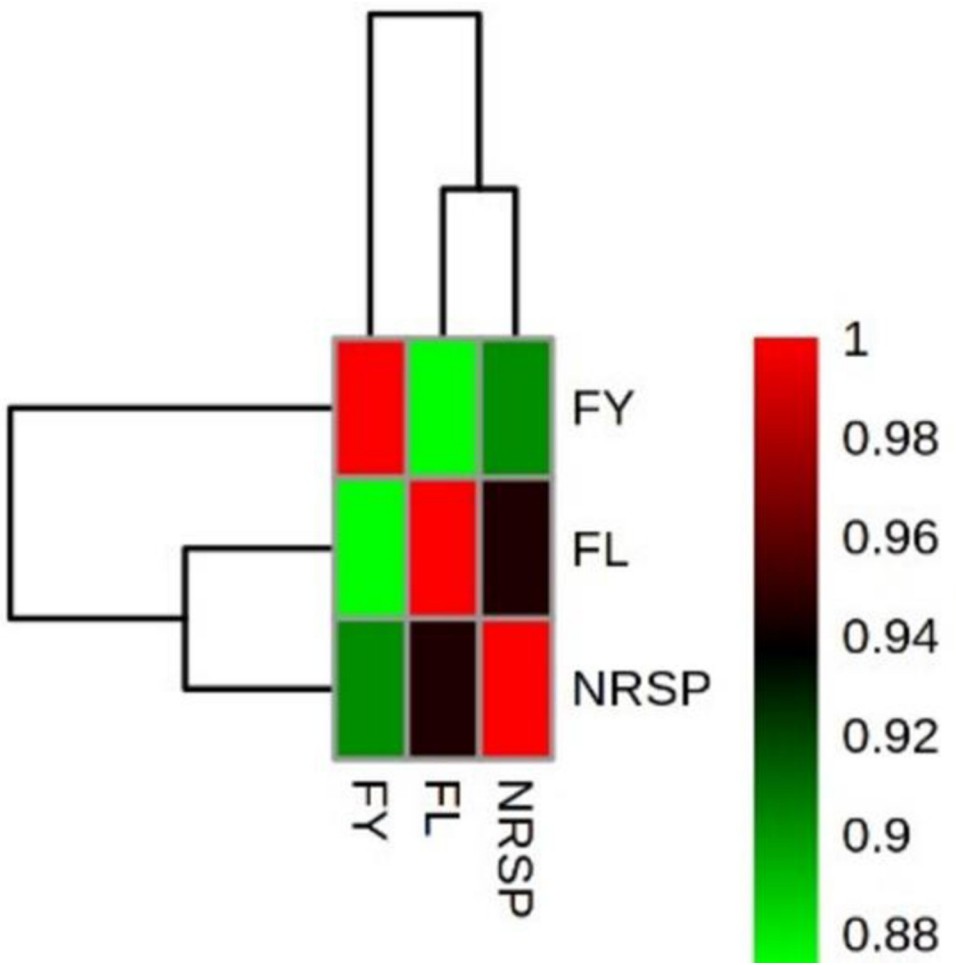

**Fig 2. Heat map correlation among the measured characters of banana.** (FY: fruit yield, FL: fruit length, and NRSP: number of rows per spike). The color scales represent the values were correlation coefficients from 0.88 (green) to 1 (red).

model for the three banana characters such as FY ($R^2$ for training: 0.92 > 0.89; testing: 0.74 > 0.65), FL (training: 0.94 > 0.91; testing: 0.90 > 0.87), and NRPS (training: 0.98 > 0.97; testing: 0.92 > 0.91) (Table 2). In addition, according to two more critical values including RMSE and MBE, the better accuracy of GRNN model in comparison with MLP was also

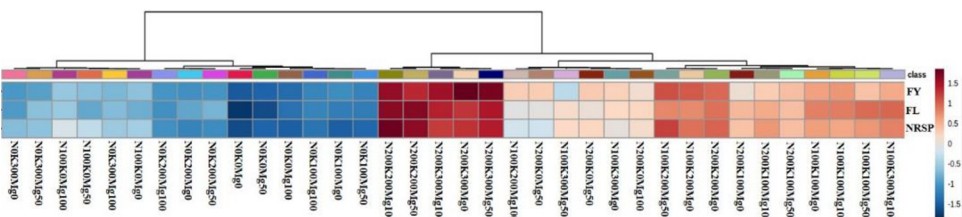

**Fig 3. Heatmap clustering of the interactions effects among the three chemical fertilizers including N (0, 100, and 200 g), K (0, 100, 200, and 300 g), and Mg (0, 50, and 100 g) on fruit yield (FY), fruit length (FL), and number of rows per spike (NRSP) of greenhouse banana.** The color scales represent the values were normalized by Z-score ((value-mean value)/standard error) for each character.

**Table 2. Performance criteria of multilayer perceptron (MLP) and generalized regression neural network (GRNN) for fruit yield (FY), Fruit length (FL), and number of rows per spike (NRPS) of greenhouse banana.**

| Model | Processing set | Performance criteria | FY | FL | NRPS |
|---|---|---|---|---|---|
| GRNN | Training | $R^2$ | 0.924964 | 0.945203 | 0.987876 |
| | | RMSE | 6.412588 | 0.832135 | 0.266243 |
| | | MBE | -1.33E-15 | -5E-16 | -2.08E-16 |
| | Testing | $R^2$ | 0.747395 | 0.903797 | 0.920342 |
| | | RMSE | 12.44093 | 1.582228 | 0.748906 |
| | | MBE | 0.509545 | -0.4 | -0.05084 |
| MLP | Training | $R^2$ | 0.895273 | 0.914649 | 0.974348 |
| | | RMSE | 7.608617 | 1.061079 | 0.387457 |
| | | MBE | -0.56688 | -0.05044 | 0.011329 |
| | Testing | $R^2$ | 0.655969 | 0.872774 | 0.919087 |
| | | RMSE | 14.87055 | 1.619764 | 0.757125 |
| | | MBE | -1.81685 | -0.49497 | -0.10114 |

$R^2$: coefficient of determination; MBE: mean bias error; RMSE: root mean square error.

confirmed. In addition, in order to find relationships between the experimental and machine learning model results, liner regression was performed. Based on the analysis a good fit correlation was observed in the experimental and machine learning model results for the FY, FL, and NRPS in the training and testing sets (Fig 4). To our best knowledge there are few studies have been used the ANNs models for predicting and optimizing the plant fertilizers. Siva [37] developed an ANN model to predict NPK fertilizers levels and recommend the best fertilizer remedy. Hartinee et al. [38] used different models such as linear, linear with plateau, quadratic, and quadratic with plateau models to determine the fertilizer recommendation rates to obtain the maximum rice yield. Macabiog et al. [39] Characterized soil nitrogen, potassium and phosphorus using ANN model. The predictive accuracy of their ANN model was confirmed in the training and testing sets with $R^2$ values of 0.998 and 0.996, respectively. Scremin et al. [24] used ANN model for modeling and predicting oat grain yield according to four levels of nitrogen as input variables.

Based on the results of the present study, the GRNN model had better performance than MLP model for modeling and predicting the FY, FL, and NRPS of banana. To our best knowledge there are no studies in the plant fertilizers area to compare GRNN and MLP models, However, some studies in other agriculture projects showed that the GRNN model was better than MLP model. Jafari and Shahsavar [40] used different ANNs models such as GRNN and MLP to predict the morphological responses of citrus to drought stress according to four input variables. The author reported that the developed GRNN had excellent efficiency for modeling the data. Hesami et al. [18] studied the effects of factors on seed germination and morphological characters of cannabis by ANNs models. The authors also reported that GRNN model revelad higher predictive accuracy than MLP model for perdicting both seed germination and morphological characters of the plant. In the current study, as the GRNN model had better performance than MLP, therefore, the model was linked to GA to optimize the FY, FL, and NRPS. Based on the results of optimization process by GRNN-GA hybrid model, the model accurately predicted that about 69, 167, and 18.6 g of N, K, and Mg would result in the maximum FY of banana outcomes, respectively (Table 3). These results confirmed that potassium is one of the most important macro element for production of banana [32, 33]. Also, according to the results of optimization process, the model accurately predicted that the amounts of the three fertilizers were almost equal for FL (64.7–75.78 g) and NRPS (55.16–61.41 g).

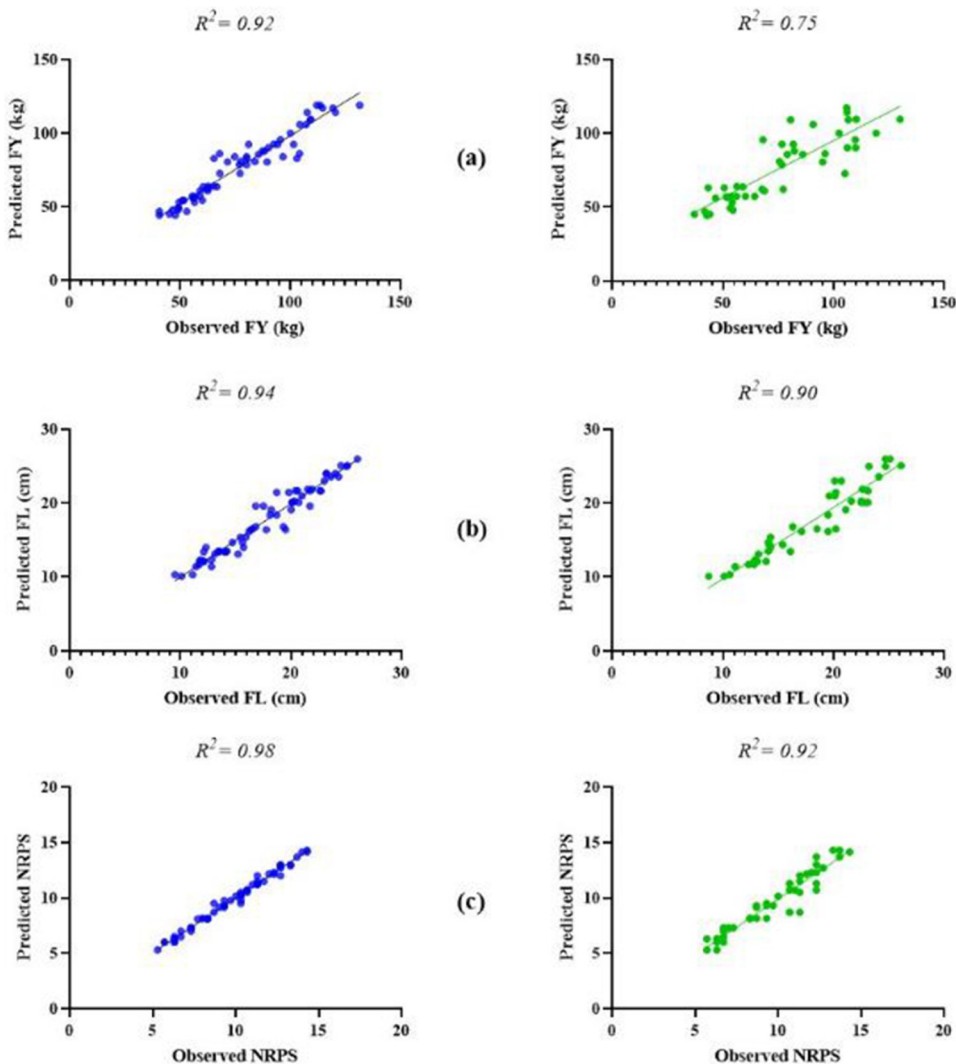

**Fig 4.** Scatter plots of the observed and predicted data of (a) FY: fruit yield, (b) FL: fruit length, and (c) NRPK: number of rows per spike of banana by GRNN model in training (blue) and testing (green) processes.

## Conclusion

According to the results the highest mean value of the FY was obtained with 200 g of N, 300 g of K, and 50 g of Mg. The GRNN model had better performance than MLP model for modeling and predicting the characters of greenhouse banana. The results of GRNN-GA hybrid model revealed that the high FY of banana can be achieved from 69, 167, and 18.6 g of N, K, and Mg, respectively. However, almost equal amounts of these elements were required to

**Table 3. Optimizing amounts of N, K, and Mg fertilizers based on genetic algorithm for maximizing fruit yield (FY), fruit length (FL), and number of rows per spike (NRPS)soybean seed yield.**

| N (g) | K (g) | Mg (g) | Prediction | Banana trait |
|---|---|---|---|---|
| 69.06 | 166.93 | 18.57 | 115.87 | FY (kg/plant) |
| 75.78 | 64.70 | 65.42 | 23 | FL (cm) |
| 53.29 | 61.41 | 55.16 | 11.5 | NRPS |

obtain high FL and NRPS. Based on the results, the GRNN-GA hybrid model can be employed in future plants fertilization studies to optimize different macro and micro-elements involved in the yield and growth of plants.

## Author Contributions

**Formal analysis:** Mostafa Farajpour.

**Software:** Mostafa Farajpour.

**Supervision:** Mahmoud Reza Ramezanpour.

**Validation:** Mostafa Farajpour.

**Writing – original draft:** Mostafa Farajpour.

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
