## [Decision Letter · Decision Letter 0]

6 Dec 2021

PONE-D-21-36053Application of artificial neural networks and genetic algorithm to predict and optimize greenhouse banana fruit yield through nitrogen, potassium and magnesiumPLOS ONE

Dear Dr.Mostafa Farajpour,

Thank you for submitting your manuscript to PLOS ONE. After careful consideration, we feel that it has merit but does not fully meet PLOS ONE’s publication criteria as it currently stands. Therefore, we invite you to submit a revised version of the manuscript that addresses the points raised during the review process.

We look forward to receiving your revised manuscript.

Kind regards,

Balasubramani Ravindran, Ph.D

Academic Editor

PLOS ONE

Journal Requirements:

" ext-link-type="uri" xlink:type="simple">https://journals.plos.org/plosone/s/file?id=ba62/PLOSOne_formatting_sample_title_authors_affiliations.pdf"

Reviewers' comments:

Reviewer's Responses to Questions

**Comments to the Author**

1. Is the manuscript technically sound, and do the data support the conclusions?

Reviewer #1: Yes

Reviewer #2: Yes

2. Has the statistical analysis been performed appropriately and rigorously? 

Reviewer #1: Yes

Reviewer #2: Yes

3. Have the authors made all data underlying the findings in their manuscript fully available?

Reviewer #1: Yes

Reviewer #2: Yes

4. Is the manuscript presented in an intelligible fashion and written in standard English?

Reviewer #1: Yes

Reviewer #2: Yes

5. Review Comments to the Author

Reviewer #1: 1. There is no need to write the complete form of the words after being introduced as abbreviated forms. Please mention the complete form on in the first appearance of the words in the main text.

2. You should mention the unit of measurement (gr etc.) for first three columns of Table 1.

3. Choose a better title for Table 2. Please pay close attention to the English language proficiency.

4. Kindly add an appropriate legend for column “prediction” in Table 3. What is the min, max and optimal number? Is there a specific range? Please provide these kind of information.

5. Please provide more in-depth caption for Fig.2 and Fig.3. Describe the correlations in Fig.2 and clustering in Fig.3 in more details.

Reviewer #2: this manuscript is an interesting work.

There are some repeated words and sentences please check them and rewrite.

please check the first paragraph of conclusion there are also repasts.

best wishes

6. PLOS authors have the option to publish the peer review history of their article (what does this mean?). If published, this will include your full peer review and any attached files.

Reviewer #1: **Yes: **Sepehr Abdolahi

Reviewer #2: **Yes: **Dr. Sakineh Abbasi

---

## [Author Response · Author response to Decision Letter 0]

14 Jan 2022

Dear Dr. Ravindran,

We would like to thank you for the letter, and the opportunity to submit a revised copy of this manuscript. We would also like to take this opportunity to express our thanks to the reviewers for the comments. We believe have resulted in an improved revised manuscript, which you will find uploaded alongside this document. The manuscript has been revised to address the reviewer comments, detailed in your letter properly as explained below.

Reviewer #1: 

1. There is no need to write the complete form of the words after being introduced as abbreviated forms. Please mention the complete form on in the first appearance of the words in the main text.

The words were written in their complete form for the first time where they were mentioned and in the rest parts we used only them as abbreviate form.

2. You should mention the unit of measurement (gr etc.) for first three columns of Table 1.

The unit of the two first characters were written in the table 1, the third characters does not unit, its number.

3. Choose a better title for Table 2. Please pay close attention to the English language proficiency.

The title of table was rewrite. 

4. Kindly add an appropriate legend for column “prediction” in Table 3. What is the min, max and optimal number? Is there a specific range? Please provide these kind of information.

To clarify we changed the title of Table 3, the values are optimal amount of the three fertilizers used in this study for the measured characters. 

5. Please provide more in-depth caption for Fig.2 and Fig.3. Describe the correlations in Fig.2 and clustering in Fig.3 in more details.

The captions of the figures were written in more in-depth.

Reviewer #2: 

A. There are some repeated words and sentences please check them and rewrite.

The sentences were rewritten 

B. please check the first paragraph of conclusion there are also repasts.

The first paragraph in conclusion section was revised.

Best regards,

Corresponding author,

---

## [Decision Letter · Decision Letter 1]

2 Feb 2022

Application of artificial neural networks and genetic algorithm to predict and optimize greenhouse banana fruit yield through nitrogen, potassium and magnesium

PONE-D-21-36053R1

Dear Dr. Farajpour,

We’re pleased to inform you that your manuscript has been judged scientifically suitable for publication and will be formally accepted for publication once it meets all outstanding technical requirements.

Kind regards,

Balasubramani Ravindran, Ph.D

Academic Editor

PLOS ONE

Additional Editor Comments (optional):

Reviewers' comments:

Reviewer's Responses to Questions

**Comments to the Author**

1. If the authors have adequately addressed your comments raised in a previous round of review and you feel that this manuscript is now acceptable for publication, you may indicate that here to bypass the “Comments to the Author” section, enter your conflict of interest statement in the “Confidential to Editor” section, and submit your "Accept" recommendation.

Reviewer #1: All comments have been addressed

Reviewer #2: (No Response)

2. Is the manuscript technically sound, and do the data support the conclusions?

Reviewer #1: Yes

Reviewer #2: Yes

3. Has the statistical analysis been performed appropriately and rigorously? 

Reviewer #1: Yes

Reviewer #2: Yes

4. Have the authors made all data underlying the findings in their manuscript fully available?

Reviewer #1: Yes

Reviewer #2: Yes

5. Is the manuscript presented in an intelligible fashion and written in standard English?

Reviewer #1: Yes

Reviewer #2: Yes

6. Review Comments to the Author

Reviewer #1: (No Response)

Reviewer #2: (No Response)

7. PLOS authors have the option to publish the peer review history of their article (what does this mean?). If published, this will include your full peer review and any attached files.

Reviewer #1: **Yes: **Sepehr Abdolahi

Reviewer #2: **Yes: **Dr. Sakineh Abbasi

---

## [Editor Report · Acceptance letter]

4 Feb 2022

PONE-D-21-36053R1 

Application of artificial neural networks and genetic algorithm to predict and optimize greenhouse banana fruit yield through nitrogen, potassium and magnesium 

Dear Dr. Farajpour:

I'm pleased to inform you that your manuscript has been deemed suitable for publication in PLOS ONE. Congratulations! Your manuscript is now with our production department. 

Kind regards, 

on behalf of

Dr. Balasubramani Ravindran 

Academic Editor

PLOS ONE